# Novel *LOX* Variants in Five Families with Aortic/Arterial Aneurysm and Dissection with Variable Connective Tissue Findings

**DOI:** 10.3390/ijms22137111

**Published:** 2021-07-01

**Authors:** Ilse Van Gucht, Alice Krebsova, Birgitte Rode Diness, Steven Laga, Dave Adlam, Marlies Kempers, Nilesh J. Samani, Tom R. Webb, Ania A. Baranowska, Lotte Van Den Heuvel, Melanie Perik, Ilse Luyckx, Nils Peeters, Pavel Votypka, Milan Macek, Josephina Meester, Lut Van Laer, Aline Verstraeten, Bart L. Loeys

**Affiliations:** 1Center of Medical Genetics, Faculty of Medicine and Health Sciences, University of Antwerp and Antwerp University Hospital, 2650 Antwerp, Belgium; ilse.vangucht@uantwerpen.be (I.V.G.); Lotte.vandenheuvel@uantwerpen.be (L.V.D.H.); Melanie.perik@uantwerpen.be (M.P.); ilse.luyckx@uantwerpen.be (I.L.); Nils.peeters@uza.be (N.P.); josephina.meester@uantwerpen.be (J.M.); lut.vanlaer@uantwerpen.be (L.V.L.); aline.verstraeten@uantwerpen.be (A.V.); 2Department of Cardiology, IKEM, Praha 4, 14021 Prague, Czech Republic; krea@ikem.cz; 3Department of Clinical Genetics, Copenhagen University Hospital, 2100 Copenhagen, Denmark; birgitte.rode.diness@regionh.dk; 4Department of Cardiac Surgery, University of Antwerp and Antwerp University Hospital, 2650 Antwerp, Belgium; steven.laga@uza.be; 5Acute and interventional Cardiology, University of Leicester, Leicester LE3 9QP, UK; da134@leicester.ac.uk (D.A.); njs@le.ac.uk (N.J.S.); tw126@le.ac.uk (T.R.W.); aab40@le.ac.uk (A.A.B.); 6Department of Human Genetics, Radboud University Medical Center, 6525 Nijmegen, The Netherlands; marlies.kempers@radboudumc.nl; 7Department of Biology and Medical Genetics, 2nd Faculty of Medicine, Charles University and Motol University Hospital, Praha 5, 15006 Prague, Czech Republic; pavel.votypka@lfmotol.cuni.cz (P.V.); milan.macek.jr@lfmotol.cuni.cz (M.M.)

**Keywords:** TAAD, *LOX*, ECM, MFS, LDS

## Abstract

Thoracic aortic aneurysm and dissection (TAAD) is a major cause of cardiovascular morbidity and mortality. Loss-of-function variants in *LOX*, encoding the extracellular matrix crosslinking enzyme lysyl oxidase, have been reported to cause familial TAAD. Using a next-generation TAAD gene panel, we identified five additional probands carrying *LOX* variants, including two missense variants affecting highly conserved amino acids in the *LOX* catalytic domain and three truncating variants. Connective tissue manifestations are apparent in a substantial fraction of the variant carriers. Some *LOX* variant carriers presented with TAAD early in life, while others had normal aortic diameters at an advanced age. Finally, we identified the first patient with spontaneous coronary artery dissection carrying a *LOX* variant. In conclusion, our data demonstrate that loss-of-function *LOX* variants cause a spectrum of aortic and arterial aneurysmal disease, often combined with connective tissue findings.

## 1. Introduction

Thoracic aortic aneurysm and dissection (TAAD) is characterized by a progressive dilatation of the aorta caused by wall weakness that leads to dissection or rupture. Unfortunately, aneurysms often remain unnoticed until dissection or rupture of the aortic wall occurs. TAAD affects thousands of people every year with a high mortality rate of almost 80%, making it a leading cause of death worldwide. In the Western population alone, TAAD is responsible for 1–2% of all deaths. More than 30 genes are known to be associated with TAAD, explaining less than 30% of all familial cases and thus leaving 70% of all families genetically unexplained [1]. With respect to non-syndromic TAAD, mutations are typically located in genes that encode for proteins involved in vascular smooth muscle cell contractility, such as *MYH11* and *ACTA2*. For syndromic TAAD forms, mutations are found in genes that encode for key players in the transforming growth factor-β (TGF-β) signaling pathway and extracellular matrix components. 

Marfan syndrome (MFS [MIM: 154700]) and Loeys-Dietz syndrome (LDS [MIM: 609192, MIM: 610168, MIM: 613795, MIM: 614816 and MIM: 615582]) are the most exhaustively studied TAAD syndromes. MFS is caused by mutations in the *FBN1* (fibrillin-1) [MIM: 134797] gene and is clinically characterized by a pleiotropy of skeletal (e.g., overgrowth), cardiovascular (e.g., TAAD and mitral valve prolapse), ocular (e.g., ectopia lentis) and skin abnormalities (e.g., striae). LDS is caused by mutations in *TGFBR1* [MIM: 190181], *TGFBR2* [MIM: 190182], *SMAD2* [MIM: 601366], *SMAD3* [MIM: 603109], *TGFB2* [MIM: 190220] or *TGFB3* [MIM: 190230] and presents with widespread arterial tortuosity and aneurysms, bifid uvula or cleft palate and hypertelorism [1,2]. The discovery of the involvement of the dysregulation of the TGF-β signaling pathway and the genetic variants that predispose to heritable aortic disease have led to novel insights [3]. Genes encoding for proteins that are involved in the assembly of the ECM also predispose to TAAD when mutated. The aortic wall must be able to withstand a lot of pressure and shear stress and therefore consists of various structured layers that guarantee the strength and elasticity of the wall.

Lysyl oxidase, encoded by the *LOX* gene, is a cuproenzyme that catalyzes crosslinking of extracellular matrix proteins such as collagen and elastin. The proper assembly of the latter key structural components of connective tissue matrices is crucial for the stability, elasticity and mechanical integrity of the aortic/arterial wall. In 2016, two publications [4,5] reported on the presence of heterozygous loss-of-function variants in the *LOX* gene in familial TAAD patients. In 2019, Renner et al. described two additional patients with *LOX* variants [6]. At the beginning of 2021, Cirnu et al. identified a missense *LOX* variant in a patient displaying arterial aneurysms throughout the body [7]. Using a next-generation sequencing approach, we aimed to validate the reported contribution of pathogenic *LOX* variants to the genetic etiology of TAAD and to further refine the associated phenotypic spectrum.

## 2. Results

We report on the identification of heterozygous *LOX* variants in five novel families, i.e., two frameshift, one nonsense and two missense variants (Figure 1A,B).

The proband of family 1 (1-III:4; Figure 2) is a 43-year-old male who presented with a type A aortic dissection (aorta ascendens diameter: 53 mm) for which he underwent aortic root and ascendens replacement. An echocardiography one year prior showed an aortic ascendens diameter of 46 mm. His clinical exam did not show specific connective tissue findings. The anatomopathological report of the resected aortic tissue showed aspecific medial cystic degeneration. The proband was known with arterial hypertension and was in follow-up because of a significant family history for aneurysmal disease. Two sisters (1-III:1 and 1-III:2) are both being followed because of ascending aortic dilatation (44 mm). His brother (1-III:6) had an aortic dissection at age 45 years. The ascending aorta diameter at the time of his Bentall surgery was 84 mm. The father (1-II:2) also underwent a Bentall surgery at age 60 and died at age 73 due to an acute myocardial infarction. A paternal uncle (1-II:3) of the proband died suddenly at age 50 years. No autopsy was performed. The son (1-III:10) of this paternal uncle is married to the sister of the proband (1-III:9). His MR angio screening at age 43 years showed an aortic sinus diameter of 43 mm (Z-score = 2.8; height 172 cm; weight 97 kg) with a sinotubular junction of 38 mm and an ascending aorta of 37 mm. Genetic testing of the proband revealed a 14 bp deletion in *LOX* (c.53_66delTAGTGCACTGCGCC) leading to a frameshift event and, as a result, the introduction of a premature stop codon (p.(Leu18Profs*111)). Segregation analysis demonstrated the presence of the *LOX* variant in the brother (1-III:6), brother-in-law (1-III:10) (who is also his nephew) and absence of the *LOX* variant in an unaffected brother (1-III:8). The sister (1-III:2; 56 years) with ascending aorta diameter of 44 mm and chronic hypertension did have a variant of unknown significance (VUS) in *FBN1* (c.5810 G > A; p.(Gly1937Glu)), but not the *LOX* mutation. The VUS in *FBN1* was also present in patient 1-III:6. DNA of two other sisters (1-III:1 and 1-III:3) (of which one (1-III:1) also has a dilated aorta) is not available. One of the two sons (1-IV:1; 25 years) of the proband also tested positive for the familial *LOX* variant. CT-scan of the entire aorta of this son (1-IV:1) revealed normal aortic diameters. The daughter (1-IV:7) of patient 1-III:10 was also found to carry the *LOX* variant but at age 16 years she still had normal aortic diameters.

The male proband of family 2 (2-III:1; Figure 2) is currently 40 years old but has a history of a type A dissection at age 19 for which he underwent composite graft surgery. He has a past medical history of spontaneous splenic rupture, spontaneous pneumothorax and early-onset varicose veins. Genetic testing showed a single nucleotide *LOX* deletion (c.351delC) leading to a frameshift event and the introduction of a premature stop codon (p.(Arg118Glyfs*119)). Familial segregation analysis showed that the *LOX* variant was also present in the father (2-II:1) and in DNA extracted from a colon polyp of the paternal grandfather. The paternal grandfather (2-I:1) lived to age 73, had a diagnosis of varicose veins, cataract, ischemic heart disease (percutaneous transluminal coronary angioplasty at age 66 years), colon polyps and rheumatoid arthritis. As an infant he was operated for an incarcerated inguinal hernia. The surgery was complicated by a subsequent infection, leaving him with a limb. At age of 30 years, he had bilateral hip replacement attributed to this. At age 50 he suffered from a contralateral inguinal hernia. The father (2-II:1) has a history of arterial hypertension. His echocardiography (at age 51 years) showed a normal aortic size and dimension of all four chambers, discrete mitral and aortic valve insufficiency and no hypertrophy. At age 45 he had an inguinal hernia, mesh repaired with good results. No other connective tissue complaints. All three affected have suffered from severe eczema.

The index patient of family 3 (3-II:5; Figure 2) is a 53-years-old male presenting with a type A dissection at age 51. He underwent an aortic valve sparing procedure according to David. He had a normal tricuspid aortic valve, but mild mitral valve regurgitation. He was negative for cardiovascular risk factors. His physical exam is only significant for tall stature (191 cm), a narrow palate and pes planus (Appendix A). His medical history includes an inguinal hernia surgery at age 40 and meniscal surgeries due to sport-related injuries. His father (3-I:1) died of lung cancer at age 73; his mother (3-I:2) lived up to age 81. He has four siblings (three brothers and one sister). Genetic testing of the index patient revealed a nonsense variant in *LOX* (c.445 G > T; p.(Gly149*)). Subsequently, two brothers (3-II:1 and 3-II:4) and a niece (3-III:1) underwent genetic testing, and all three were tested positive for the familial *LOX* variant. Cardiac CT-scan of the 56-years-old brother (3-II:4) demonstrated a borderline aortic sinus measurement of 41 mm (Z-score = 2.3; height: 199 cm; weight: 104 kg), but normal ascending aorta. From his medical history we retain an inguinal hernia surgery at age 3, a history of hypertension (treated with metoprolol), hypothyroidy (age 40 years), rupture of the supraspinatus ligament (age 41 years) and tibialis posterior neuralgia (age 42 years). The other brother (3-II:1; 59 years) also has a history of bilateral inguinal hernia, tall stature and joint hypermobility. His echocardiographic evaluation at age 59 years showed no evidence of aortic dilatation. His daughter (3-III:1; 28 years) underwent “urgent” pre-symptomatic genetic testing because of pregnancy and tested positive. She is known with joint hypermobility, arachnodactyly, pes planus, shoulder dislocations and multiple ankle distortions. Her echocardiography revealed normal aortic dimensions (sinus of Valsalva 28.6 mm; Z-score = 0.28). She is known with hypothyroidy (age 19 years), gastric sleeve surgery (25 years), cesarian section (27 years) with premature delivery (26 weeks) of male newborn who died at one week of age.

This female proband (age 50) of family 4 (4-II:1; Figure 2) has a history of postero-lateral myocardial infarction due to spontaneous coronary artery dissection at age 44 and a left internal carotid artery dissection at age 46. She has a history of fibromuscular dysplasia of the renal arteries, increased skin elasticity, generalized Beighton score of 4/9, enlarged armspan (167.5 cm; ratio 1.08) and recurrent episodes of ankle dislocation (Appendix A). Wound healing was normal. Echocardiography revealed a diameter of 35 mm at the level of the sinuses of Valsalva (Z-score = 0.8 for height 154.6 cm and weight 93.9 kg), 26 mm at the sinotubular junction and 30 mm for the ascending aorta. Genetic testing revealed a *LOX* missense variant (c.893T > G; p.(Met298Arg)) which was previously reported as pathogenic in an unrelated family [4]. Her family history reveals her mother (4-I:2) died at age 58 due to an intracranial bleed of a berry aneurysm but her DNA sample is not available for testing. The father (4-I:1) and brother (4-II:2) died at, respectively, 58 and 40 years old due to acute myocardial infarction; both were known with significant cardiovascular risk factors. Another brother (4-II:3; age 54) is known with chronic obstructive pulmonary disease (COPD) and no DNA is available from him. 

The male index patient of family 5 (5-III:1; Figure 2) (21 years) was initially diagnosed with aortic dilatation (both at the level of the sinuses and even more pronounced at the ascending aorta) at the age of 6. At the age of 14, he underwent elective replacement of the ascending aorta because of fast progression (10 mm per 2 years) of its diameter to a maximum of 50 mm at the sinotubular junction and above. Microscopic findings showed an abnormal texture of the elastic component of the vascular adventitia, with suspicion of a connective tissue disorder. His height is at the 95th percentile (Appendix A). There is no characteristic facial dysmorphism or joint hypermobility and his skin has normal elasticity. His father (5-II:1) committed suicide. Nonetheless, aortic dissection has been reported in the father’s family, where the paternal grandfather (5-I:2) and his brother (5-I:1) died of acute aortic dissection before the age of 50 years. The mother (5-II:2) was shown to have a normal echocardiography. Mutation analysis revealed the presence of a *LOX* missense variant (c.917T > C; p.(Leu306Pro)), which was absent from the unaffected mother.

According to the ACMG criteria, four out of five *LOX* variants are predicted to be pathogenic (Appendix A), whereas variant p.(Leu306Pro) remains a variant of uncertain significance. All variants are absent from gnoMAD database. Three variants create premature stop codons (c.53_66delTAGTGCACTGCGCC; p.(Leu18Profs*111); c.351delC; p.(Arg118Glyfs*119) and c.445 G > T; p.(Gly149*)), which are predicted to result in nonsense-mediated mRNA decay. The *LOX* missense variants p.(Met298Arg) and p.(Leu306Pro) are located in the catalytic domain of *LOX* (Figure 1A), located at highly conserved amino *LOX* acid positions (Figure 1B) and are predicted to be disease-causing by different prediction programs (Appendix A). The first one (p.(Met298Arg)) has been previously reported in a TAAD family and has been proven to affect collagen-elastin crosslinking and, hence, aortic wall integrity in the respective mouse model [4].

Histological staining of aortic wall sections of *LOX* patients (1-III:4 and 1-III:6) with Verhoeff-Van Gieson staining (VVG) revealed increased elastic fiber fragmentation and disorganization in comparison to a healthy control (Figure 3). Masson’s Trichrome staining to assess the collagen content in the same aortic tissue samples showed variable collagen content in the patients. To investigate the involvement of the TGF-β signaling pathway in *LOX* deficiency and TAA development, immunohistochemistry was used to determine the levels of nuclear pSMAD2 and pERK1/2 in the aortic segments of the *LOX* patients. Compared to a control, a larger fraction of pSMAD2 and pERK1/2 positively stained nuclei can be observed in aortic tissue of both *LOX* patients (Figure 4).

## 3. Discussion

In 2016, (likely) pathogenic variants in *LOX* were described as a genetic cause for arterial aneurysm and dissection [4,5]. From the nine initially described variants, three were premature stop codons (one recurrent p.(Trp42*)) and all six missense variants were located in the LOX catalytic domain (Figure 1A) [4,5,6,7]. We here describe five additional variants, including three nucleotide deletions or substitutions resulting in a premature stop codon and two missense variants affecting highly conserved amino acids (Figure 1B) in the LOX catalytic domain. Overall, the variant pattern confirms loss-of-function as the underlying cause. 

Comparison of the yet existing literature on TAAD-causing *LOX* variants with our own series of patients reveals a number of interesting observations. A summary of the clinical findings of our cohort of *LOX* patients and *LOX* patients in literature can be found in Table 1. A more detailed overview of the clinical features from literature can be found in Appendix A.

First, with regard to the demographic characteristics of the patient population from literature, there is a wide range in presenting ages from 11 to 73 years (median 46 years) for cardiovascular findings. In our patient cohort, the earliest aortic aneurysm observation occurred at age 6 and the first thoracic aortic dissection occurred at age 19. In other families, we also observed non-penetrance at age 51 years (2-II:1) and 59 years (3-II:1) in family 2 and 3, respectively. As suggested in the prior LOX reports [4,5,7], there is a predilection for male gender. Overall, 77% (33/44) of the *LOX* patient population is male. The observation of male predilection with more severely dilation and more frequent dissection in males has been observed in other TAAD conditions as well, both in mice and humans [8].

Second, the *LOX* variants seem to mainly cause aneurysms and dissections at the level of the thoracic aorta. Overall, eight aortic dissections that have been described, all affecting the ascending thoracic aorta, except for one that starts in the aortic arch. Thoracic aortic dissection seems to occur after significant aortic enlargement, with the largest aneurysm reaching an impressive 125 mm in the ascending aorta. So far, no type B or abdominal dissections have been described. However, a patient with abdominal aortic aneurysm has been reported [7]. Arterial disease beyond the ascending aorta seems less frequent, with an extension of thoracic aortic aneurysm into the brachiocephalic artery, one hepatic arterial aneurysm and only one patient displaying aneurysm throughout the body in literature [4,7]. We were the first to report spontaneous coronary artery dissection as part of the *LOX* phenotype [9]. We observed no difference between premature stop codon *LOX* variants (*n* = 23) and missense *LOX* variants (*n* = 22) with regards to median age of diagnosis (47.5 years versus 44 years) or severity of the cardiovascular involvement.

Third, extra-aortic cardiovascular features include bicuspid aortic valve disease (3/18 in initial series [5]), as well as mitral and aortic valve insufficiency (*n* = 3). Arterial tortuosity has been mentioned once but was still within the normal age-related variation. In our own patient cohort, we did not observe bicuspid aortic valve disease, whereas mitral and aortic valve insufficiency were observed once. 

Fourth, several patients in the literature (*n* = 10) and in our series (*n* = 9) have been reported to present with other connective tissue findings. These include tall stature (7/19), pectus deformities (4/19), scoliosis (2/19), joint hypermobility with dislocation (8/19), inguinal hernia (5/19), skin striae (6/19) and dural ectasia (3/19) [4,5,6,7]. As such, there is significant clinical overlap with other syndromic aortic aneurysmal conditions such as Marfan syndrome and Loeys-Dietz syndrome [2]. Remarkably, one of our patients (proband of family 2) also presented with splenic rupture, a finding also described in vascular Ehlers-Danlos syndrome and Loeys-Dietz syndrome [2]. We also for the first time describe the occurrence of spontaneous pneumothorax, a clinical complication well known to present in several connective tissue disorders [10]. Varicose veins have also been reported in at least two families (Lee et al., 2016 [4] and family 2 in our cohort). No connective tissue findings seem specific for *LOX* mutation carriers, and it also remains unclear at present why some *LOX* mutation carriers do not seem to present with connective tissue findings at all.

Fifth, histopathological findings of the aorta seem rather aspecific, with elastic fiber fragmentation and cystic medial degeneration. These anatomopathological observations are a common theme in aortic aneurysmal walls of patients with diverse connective tissue disorders, but do not allow prediction of the underlying gene defect. Immunohistological stainings of TGF-β pathway key effectors, pSMAD2 and pERK1/2, revealed an increase of positively stained nuclei, an indication for increased TGB-β signaling, in aortic tissue sections from *LOX* patients. Our observations confirm prior hypothesis that enhanced TGF-β signaling can contribute to TAA development caused by LOX deficiency [11].

## 4. Materials and Methods

The cohort consists of genetically undiagnosed patients (*n* = 16) with aortic or arterial aneurysm and dissection belonging to 5 families, both in the presence and absence of connective tissue findings. 

Next-generation TAAD gene panel (Appendix A) sequencing was performed on DNA of probands according to previously described methods [12]. In brief, enrichment of the regions of interest was performed with a custom Haloplex enrichment kit according to the manufacturer’s protocol (Agilent Technologies). The concentration of each library was measured by Qubit fluorometric quantification (Life Technologies). For generation of clusters and subsequent sequencing of the targeted DNA samples on a flow cell, a sequencing reagent kit from Illumina was used. High-throughput next generation sequencing data were generated on an Illumina platform. Analysis of the raw data was performed using Seqpilot (JSI) or a Galaxy pipeline, followed by variant calling with the Genome Analysis Toolkit (GATK) and variant annotation using an in-house developed tool, VariantDB [13]. Identified variants were genotyped in both affected and unaffected available family members using Sanger sequencing.

Histological examination was performed on aortic wall tissue from two of our patients (Family 1 (p.(Leu18Profs*111)), 1-III:4 and 1-III:6) and one control that was collected during surgery. The collected tissue was embedded in paraffin and sections of 5μm thick were created. Verhoeff Van Gieson (Sigma-Aldrich) staining was performed to assess elastic fiber integrity, while Trichrome Masson’s (Sigma-Aldrich) staining was used to evaluate the collagen content. pSMAD2 and pERK1/2, important components of the TGF-b signaling pathway, were visualized using immunohistochemistry. Sections were deparaffinized in toluene for 5 min and rehydrated in alcohol grades of 100%, 90%, 70% and 50%. Sections were submerged in 3% hydrogen peroxide to inhibit endogenous peroxidase, followed by 10 min in trypsin at 37 °C (Sigma-Aldrich, 93615-25G) for antigen retrieval and boiled in citrate buffer in a microwave for 3 s at 650 W and 10 min at 90 W. Sections were cooled down at room temperature for 30 min, rinsed with MiliQ and TBST and blocked for 20 min at room temperature with goat serum (Vector-lab consult). A primary antibody concentration of 1:5000 and 1:3000 for pSMAD2 (3101, Cell Signaling) and pERK1/2 (4370, Cell Signaling), respectively, was used to incubate the sections in a humid chamber at room temperature overnight. The day after, sections were rinsed in TBST followed by incubation with the secondary antibody (1:200, 30014 Vector, Sec goat anti-rabbit IgG) for 30 min at RT and with the avidin-biotinylated complex for 1 h at RT (Vectastain ABC kit, Vector Laboratories). 3,3-Diaminobenzidine tetrahydrochloride hydrate (DAB) chromogen (Sigma-Aldrich) was used as the substrate and Hematoxylin as counterstaining. An automated Nikon Ti-E inverted microscope equipped with a Nikon DS-U3 digital camera (Nikon Instruments) was used to acquire staining images. Nikon NIS Elements AR software v4.51.01 was used for image acquisition at 10× magnification.

## 5. Conclusions

Taken together, our data suggest that loss-of-function *LOX* variants cause a wide spectrum of aortic and arterial aneurysmal disease, combined with connective tissue findings such as inguinal hernia, pneumothorax, varices, joint dislocations and splenic rupture. The time of onset of aortic aneurysm seems variable but can be as early as 6 years of age. We also report the occurrence of coronary artery dissection as part of the *LOX* phenotype.

## Figures and Tables

**Figure 1 ijms-22-07111-f001:**
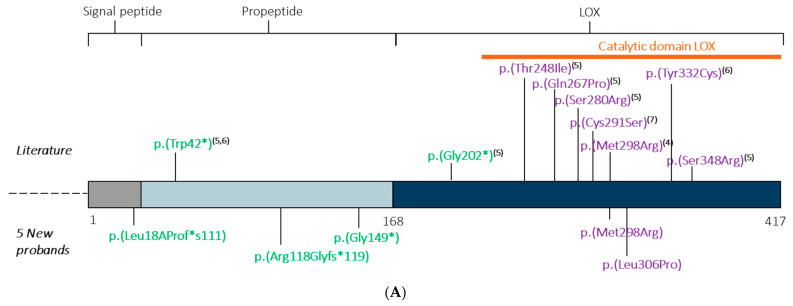
(**A**) Overview LOX protein and location of variants found in five new probands and literature. Variants leading to premature stop codons and missense variants are highlighted in, respectively, green and purple [4,5,6,7]. (**B**) Conservation of affected LOX amino acids.

**Figure 2 ijms-22-07111-f002:**
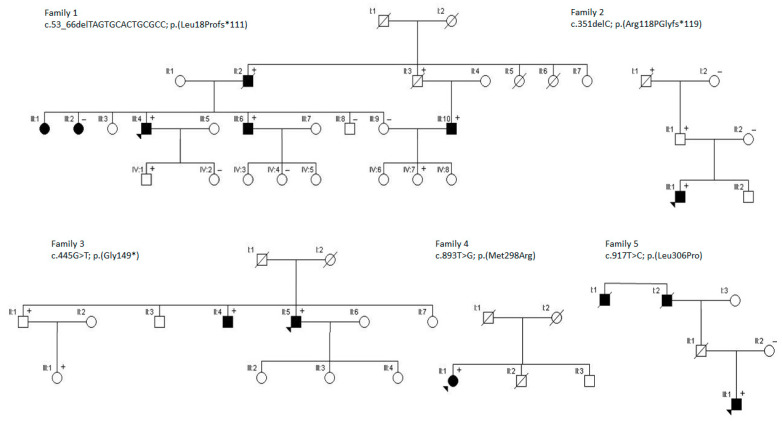
Pedigrees of *LOX* families. The probands are indicated with an arrow. Black symbols indicate family members that are affected with aortic/arterial aneurysm/dissection. Deceased individuals are indicated with a diagonal line. Plus and minus signs are used to indicate the presence and absence of the respective *LOX* variant.

**Figure 3 ijms-22-07111-f003:**
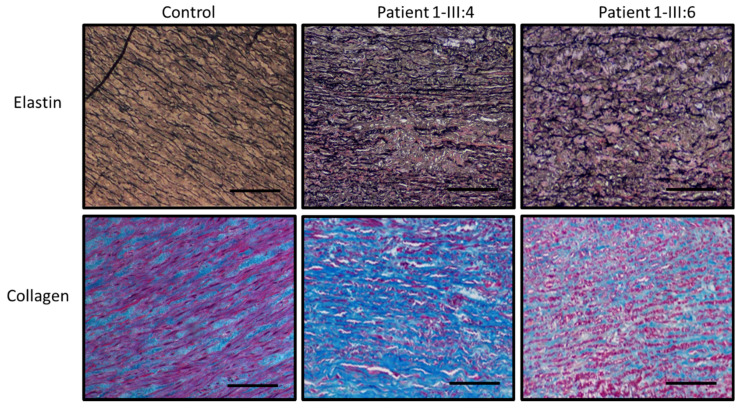
Representative pictures of collagen and elastin stainings performed in a control and two *LOX* patients, respectively, patient 1-III:4 and 1-III:6. The scale bar represents 100 μm.

**Figure 4 ijms-22-07111-f004:**
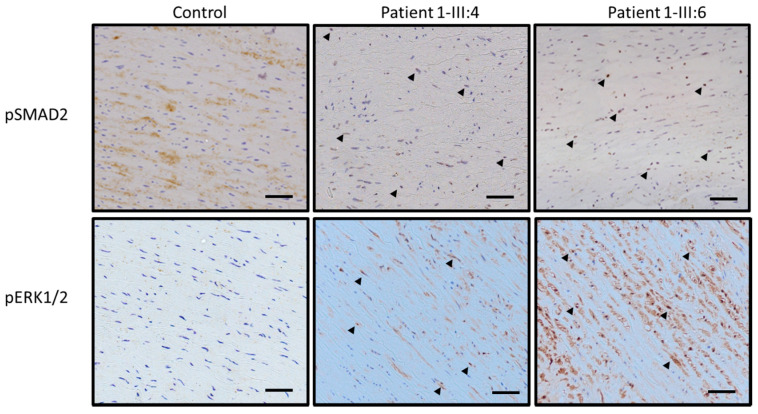
Pictures of immunohistochemistry stainings of pSMAD2 and pERK1/2 in a control and two *LOX* patients, respectively, 1-III:4 and 1-III:6. Black arrows indicate examples of positive stained nuclei. The scale bar represents 50 μm.

**Table 1 ijms-22-07111-t001:** Summary of clinical findings of our cohort of patients with *LOX* variants compared to literature.

Characteristics	Literature [4,5,6,7]	Our Cohort
Total number of patients	*n* = 29	*n* = 16
Male/female	20/8 (1 unknown)	13/3
Age range at diagnosis	11–73 years (median 46 years)	6–55 years (median 45 years)
Thoracic aortic dissections	*n* = 3 ascending, *n* = 1 arch	*n* = 4 ascending
Elective surgeries	*n* = 13 (age 11–70 years)	*n* = 1 (age 14 years)
BAV	*n* = 3	*n* = 0
Disease beyond ascending aorta	Aneurysm in brachiocephalicus, arteria hepatica, abdominal aorta Arterial tortuosity Coronary artery aneurysm	Coronary artery dissection
Connective tissue findings	*n* = 10	*n* = 9

## Data Availability

The data presented in this study are openly available and submitted to ClinVar (https://www.ncbi.nlm.nih.gov/clinvar/) (Genbank: NM_002317.7; accession numbers SCV001623447, SCV001623448, SCV001623449, SCV001623450, SCV001623451).

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
