# Peer review of "Novel LOX Variants in Five Families with Aortic/Arterial Aneurysm and Dissection with Variable Connective Tissue Findings"

_ijms, 2021, doi:10.3390/ijms22137111_

Round 1

Reviewer 1 Report

Van Gucht and colleagues map the occurrence of loss-of-function variants within the LOX gene in five families with thoracic aortic/ arterial aneurysms using the NGS TAAD gene panel. 
1. Could you please specify or hypothesize based on current findings which LOX variant(s) may be connected with the development of TAAD early in life?
2. Why there was histological examination performed only in two probands with the same LOX variant from family 1?
3. It should be interesting to perform RNASeq in aortic tissue to complement your results.

Author Response

Answers to questions/comments of reviewer 1

Van Gucht and colleagues map the occurrence of loss-of-function variants within the LOX gene in five families with thoracic aortic/arterial aneurysms using the NGS TAAD gene panel. 

  1. Could you please specify or hypothesize based on current findings which LOX variant(s) may be connected with the development of TAAD early in life?

As suggested by the reviewer we performed a comparative analysis of the clinical phenotype premature stop LOX variants (n=22) and missense LOX variants (n=23). We did not observe a difference with regards to median age of onset (47.5 yrs versus 44 yrs) or severity of the cardiovascular involvement. Our cohort is probably still too small to observe a difference. We added an additional sentence on this comparison to the main text.

  1. Why there was histological examination performed only in two probands with the same LOX variant from family 1?

The stainings were only performed on those two individuals because these were the only paraffin embedded aortic tissues that were available for histological evaluation.

  1. It should be interesting to perform RNASeq in aortic tissue to complement your results.

We agree with the reviewer that this would be an interesting experiment, however at present we do not have access to frozen or fresh aortic tissue to perform this RNAsequencing. If in the future we would have access to a patient undergoing surgery, we will preserve such material and execute RNAseq.

Reviewer 2 Report

The paper is a useful addition to the currently limited literature on aortic aneurysm associated LOX gene variants. It describes an additional 16 individuals from 5 families, significantly increasing knowledge of associated  clinical phenotypes associated with LOX variants. The paper expands on the cardiac findings, specifically that of coronary artery dissection,  in addition to a range of other connective tissue phenotypes. The paper summarises clinical and molecular data from previously reported literature cases. The authors report  histopathological findings of aortic wall samples from 2 variant carrier patients , confirming previously reported fragmentation of elastin deposition and adding analysis of TGF-b signaling pathway components.

Strengths include a clearly structured and well written article and the data summarised in the discussion session.

The 3 main previous references are included (Guo, Lee, Cirnu) but I would suggest adding Renner et al Genetics in medicine 2019 (DOI 10.1038/s42436-019-0435-z) which includes the Trp42 variant and a VUS p.Tyr332Cys associated with additional connective tissue findings.

The reference numbers in figure 1 of previously described and novel variants is incorrect (refs 4 and 5 are switched). 

It is not immediately obvious to me why figure 1 includes p.Leu154Phe (Guo et al) which was deemed likely benign and is listed in clinvar as such. I this this would come out as a cold VUS by ACMG criteria however I am unable to see any supplemental data (the only supplemental file available to me was the authors covering submission letter). It would be worth commenting on this if not already covered there. 

Author Response

Answers to questions/comments of reviewer 2.

The paper is a useful addition to the currently limited literature on aortic aneurysm associated LOX gene variants. It describes an additional 16 individuals from 5 families, significantly increasing knowledge of associated  clinical phenotypes associated with LOX variants. The paper expands on the cardiac findings, specifically that of coronary artery dissection,  in addition to a range of other connective tissue phenotypes. The paper summarises clinical and molecular data from previously reported literature cases. The authors report  histopathological findings of aortic wall samples from 2 variant carrier patients , confirming previously reported fragmentation of elastin deposition and adding analysis of TGF-b signaling pathway components.
Strengths include a clearly structured and well written article and the data summarised in the discussion session.

Thanks for the encouraging and positive comments to our paper.

The 3 main previous references are included (Guo, Lee, Cirnu) but I would suggest adding Renner et al Genetics in medicine 2019 (DOI 10.1038/s42436-019-0435-z) which includes the Trp42 variant and a VUS p.Tyr332Cys associated with additional connective tissue findings.
We have added both variants to the Figure 1 and also included their associated phenotypes in the Table 1 and Table S3.

The reference numbers in figure 1 of previously described and novel variants is incorrect (refs 4 and 5 are switched). 
Thanks for pointing at this switch. We have corrected the references.

It is not immediately obvious to me why figure 1 includes p.Leu154Phe (Guo et al) which was deemed likely benign and is listed in clinvar as such. This this would come out as a cold VUS by ACMG criteria however I am unable to see any supplemental data (the only supplemental file available to me was the authors covering submission letter). It would be worth commenting on this if not already covered there. 
We agree that this variant is most likely benign and we have removed it from Figure1.